# Design and Implementation of an Ultra-Wideband Water Immersion Antenna for Underwater Ultrasonic Sensing in Microwave-Induced Thermoacoustic Tomography

**DOI:** 10.3390/s24196311

**Published:** 2024-09-29

**Authors:** Feifei Tan, Haishi Wang

**Affiliations:** College of Communication Engineering (College of Microelectronics), Chengdu University of Information Technology, Chengdu 610225, China

**Keywords:** ultra-wideband antenna, ultrasonic sensor, biomedical imaging, MITAT

## Abstract

Microwave-induced thermoacoustic tomography (MITAT) holds significant promise in biomedical applications. It creates images using ultrasonic sensors to detect thermoacoustic signals induced by microwaves. The key to generating thermoacoustic signals that accurately reflect the fact is to achieve sufficient and uniform microwave power absorption of the testing target, which is closely tied to the microwave illumination provided by the antenna. In this article, we introduce a novel design and implementation of an ultra-wideband water immersion antenna for an MITAT system. We analyze and compare the advantages of selecting water as the background medium. Simulations are conducted to analyze the ultra-wideband characteristics in impedance matching, axial ratio, and radiation pattern of the proposed antenna. The measured |S_11_| shows good agreement with the simulated results. We also simulate the microwave power absorption of tumor and brain tissue, and the uniform microwave power absorption and high contrast between the tumor and brain indicate the excellent performance of the proposed antenna in the MITAT system.

## 1. Introduction

Cancer is a large group of diseases that starts in almost any organ or tissue of the body, and the key to improving cancer outcomes and cancer survival is early detection [1,2]. Various methods have been evaluated as cancer screening tools. Among these, medical imaging technology occupies an important place [3,4,5,6,7]. However, different imaging methods are appropriate for different medical needs. X-ray computed tomography (CT) is mature and widely used, but its ionizing radiation makes it unsuitable for broad screening and frequent use [8,9]. Magnetic resonance imaging (MRI) is non-ionizing and obtains clearer images of soft tissues such as the brain, ovary, and liver. However, it is time-consuming and expensive, which brings limitations to routine cancer screening [10,11]. Ultrasound imaging (UT) is one of the conventional medical examination methods in present hospitals [12,13]. It is non-ionizing, has a low cost, and has a high resolution. However, it does not apply to tumor cell detection when the acoustic impedance difference between a tumor and the surrounding tissue is small, such as the tumor in the brain and breast. Microwave imaging (MWT) is an emerging research field that avoids using ionizing radiation and would be inexpensive, quick, and comfortable for patients [14,15]. When the human body is illuminated with microwave radiation, tumors, and the surrounding tissues induce different reflections and scatterings due to the differences in dielectric properties [16]. However, two main challenges of low resolution and difficulty in identifying received signals from same-frequency transmitted signals have limited its use.

Based on the interdisciplinary research of electromagnetics, acoustics, biology, and medical science, microwave-induced thermoacoustic tomography (MITAT) has become another potential alternative technique for medical imaging [17,18]. The physical mechanism is based on the thermoacoustic phenomenon. When the tissue is illuminated by high-peak-power microwave pulses, it absorbs microwave energy and generates microwave-induced thermoacoustic signals (TA signals) due to thermoelastic expansion. The TA pressure amplitude PMIT is proportional to the quantity of the absorbed microwave power PMW, and PMW has a functional relationship with the electrical property of the tissue. The relationship between PMIT and PMW is shown in Equation (1), and α is the attenuation coefficient. When the testing target contains different kinds of tissues, such as tumors and the surrounding healthy tissues, it generates different values of PMIT. After that, an ultrasonic sensor runs a full scan of electric machinery and transmits the data to the computer to complete data processing; then, the microwave power absorption distribution map is reconstructed, which is the image of the testing target. The detection of ultrasonic signals is also run by a sensor array [19].
(1)$$P_{MIT}\propto 2\alpha P_{MW}$$

The block diagram of the whole process of the MITAT system is shown in Figure 1. The system is divided into two parts: the MW process and the UT process. The former contributes similar high contrast with MWT due to the large difference in dielectric conductivity between a tumor and the surrounding tissue, and the latter contributes high resolution with UT. The integration of the advantages of MWT and UT extends the application range of MITAT.

The data source of computer processing and imaging is obtained from ultrasonic sensor detection, whose accuracy and reliability determine the image quality [20]. One of the solutions to obtain satisfied sensor detection results is to improve the accuracy of the microwave power absorption condition of the target, which is closely related to the microwave illumination from the antenna.

Different types of antennas are designed and used in the MITAT systems, and the comparison with related antennas in the microwave-induced thermoacoustic (MITAT) systems is listed in Table 1. Yan, A. et al. proposed a quasi-conical spiral antenna with circularly polarized radiation characteristics. It can achieve a uniform distribution over a large imaging domain with a diameter of 10 cm [21]. The helical antenna proposed in reference [22] has a similar structure to that in reference [21] and also has good circular polarization radiation characteristics. As a comparison, the customized horn antenna was used to provide linearly polarized illumination in reference [21]. Based on the simulated and experimental results, the images under circularly polarized illumination presented better quality than those under linearly polarized illumination. In addition, He, Y. et al. promoted the research by using an implementation of a circularly polarized illumination method in the MITAT [23]. The key step is to insert a single-layer linear-to-circular polarizer based on frequency-selective surfaces between a pyramidal horn antenna and an imaging object. According to Table 1, the traditional device to import microwave is the waveguide [24,25], which is usually linearly polarized. According to the studies mentioned above, the microwave transmitting method by the polarizer makes it more suitable for a waveguide-based MITAT system. Moreover, the design goal of circular polarization of the proposed antenna in this work also benefited from the studies in [22]. 

In this work, in order to meet the requirements mentioned above, the following three issues are concerned with the design of the proposed antenna. Firstly, the core circuits are designed using microstrip lines with the advantages of low profile, low fabrication cost, high radiation efficiency, and ease of manufacturing. Secondly, the antenna is circularly polarized. A common problem in the MITAT is the nonuniform microwave power absorption resulting from the inhomogeneous microwave illumination [28,29,30]. Using a circularly polarized antenna has been demonstrated to be useful in achieving more uniform microwave power absorption in tumors by Yu He et al. [22], thereby reducing the adverse effect of inhomogeneous illumination. Lastly, the antenna is operating underwater. As shown in Figure 1, microwave and ultrasonic waves flow in the system simultaneously, and water is the conditioner that decreases the power propagation losses and return losses of both. Other materials may also be used as the background medium, but water has the advantage of lower power loss and is readily available. 

In recent years, underwater wireless communication has drawn the attention of the research community due to its broad applications [31], including underwater wireless sensor networks [32], remote-operated vehicles, and diver-to-diver communications. To develop reliable and efficient underwater communication, significant effort has been devoted to the designing of water immersion antennas [33,34,35]. Most of the reported antennas are electrically small, which either have low radiation efficiency due to high ohmic loss or use inefficient nonresonant size that leads to huge power consumption [36,37]. The magnetoelectric antennas that radiate electromagnetic waves by physically moving, rotating, or oscillating the electric charges or magnetic dipole moments have provided a possible way to miniaturize the antenna [38,39,40,41], but the problems of power inefficient and not easily packaged in water could not meet the challenge of high peak power radiation in the MITAT system [42,43,44]. 

The comparison of the outcomes of the antennas applied in similar application scenarios has been made and is shown in Table 2. The antenna in reference [25] and this work both use water as the background medium, and the size of the proposed antenna is far less than the other. Compared with other antennas, the size of the proposed antenna is in middle or low levels. The wide-slot antenna in reference [45] has a smaller size. It is designed with microstrip lines, which is the same as in this work. However, it is applied in the microwave tomography (MWT) system, which only uses a microwave for imaging. The background medium is made of oil of paraffin and distilled water that is friendly with the microwave. For this work, the background medium must be friendly to both microwave and acoustic waves at the same time. Therefore, there are more challenges in antenna design.

Evangelia A. Karagianni proposed a bow-tie microstrip antenna for a wireless sensor network (WSN) that is used to monitor environmental or physical phenomena of the sea and cooperatively disseminate the data through the network of sensors to a shore access point [46]. The antenna is omnidirectional with a gain of 1.2 dBi. The input reflection coefficients under seawater are −12 dB and −16 dB at the S and C bands, respectively. Majid Ostadrahimi et al. made an array that contains 24 dipole elements for microwave imaging [47]. The impedance matching condition is optimized by adjusting the salinity of the background medium. David Gibbins et al. proposed a wide-slot UWB antenna for an MWT system [45], of which the background medium is specially prepared and has similar dielectric properties with human breast tissue in order to reduce reflections by eliminating the air/skin interface. However, this leads to additional preparation for experiments and limits the application of the system.

**Table 2 sensors-24-06311-t002:** The comparison of the outcomes of novel antennas applied in MITAT and MWT systems.

Reference	Antenna	Operation Frequency/ Bandwidth	Size/Aperture	Application Scenario	Background Medium
[21]	Quasi-conical spiral antenna	3 GHz	17.5 × 182 mm^2^	MITAT	Mineral oil
[22]	Horn antenna	3 GHz	476 × 346 × 700 mm	MITAT	Mineral oil
[23]	Polarized antenna	3 GHz	175 × 156 mm^2^	MITAT	Mineral oil
[27]	Beamforming array antenna	1.6–3.3 GHz	Circle with radius of 50 mm	MITAT	Mineral oil
[24]	Waveguide	2.45 GHz	109.22 × 54.61 mm^2^	MITAT	Acoustic coupling
[25]	Cascaded Waveguide	3 GHz	72 mm × 34 mm plus 72 mm × 5 mm	MITAT	Water
[47]	dipole antenna array	0.8–1.2 GHz	Circle with radius of 223.5 mm	MWT	Salt water
[45]	Wide-slot antenna	About 4.3–5.7 GHz; 6.5–10 GHz	14 mm × 13 mm × 1.25 mm	MWT	Customed Medium
This work	Water-immersion antenna	0.84–2.75 GHz	55 mm × 55 mm × 63.2 mm	MITAT	Water

Wideband performance in the antenna makes it a potential candidate for more applications, such as portable MITAT systems, WLAN, WiFi, and C band applications [48]. By the application of fractal geometry in antenna engineering, the fractal antenna can obtain miniaturization and wideband characteristics [49,50,51]. The compact, flexible antenna also has the advantages of compact size and ultra-wide bandwidth, which are easier to integrate into other systems [52,53]. Shaw, T. et al. proposed a compact wideband implantable flexible antenna by modeling the entire device on a thin dielectric substrate [54]. To attain biocompatibility with human tissue and reduce the antenna profile, a biocompatible coating layer of alumina (Al_2_O_3_) is considered all around the implantable CP antenna.

With recent advances in III-V semiconductor RF power sources [55], high-performance transducers [56], advanced signal processing techniques [57], and flexible materials [58], the MITAT has the potential to be integrated into a portable system.

Qin, H. et al. proposed a dipole antenna for a homemade MITAT system. The notable feature of the antenna is that it is simple and small, which can be easily integrated with the system and is portable [26]. However, the feasibility of the system in detecting small pancreatic tumors is due to the intravenous infusion of anti-Gal1-Fe_3_O_4_ nanoparticles. Nan, H. et al. proposed a beamforming array to focus the RF energy at the target location and can enable lower peak RF power levels, providing a pathway to miniaturization [27].

These types of antennas feature good bandwidth and a simple fabrication process. However, the lower power capacities, limited gains, and relatively high cost restrict the application in the MITAT system.

This article proposed a water immersion antenna for the MITAT system that features a simple structure, small size, easy fabrication, ultra-wide bandwidth, circular polarization, and low cost, which is suitable to be applied in the MITAT system.

## 2. Power Loss Analysis

The special feature of the MITAT system is that there are two kinds of waves in the background medium when it is imaging, as shown in Figure 1. To simplify the analysis, we ignore the power losses induced by the antenna radiation angle, as well as some other power losses, like the possible impurities in the background medium and the scattering. Thus, the power losses of both waves occur only in two ways: the power transmission loss in the background medium and the reflection loss on the interface of the tissue.

The background medium should be friendly to both waves, which means the power losses related to it must be low. This work has made comparisons of four background media and four biological tissues, and the relevant acoustic and electric parameters are shown in Table 3. Water and oil are two common background media in the MITAT system, according to Table 1. Milk has a similar acoustic impedance to breast fat. The customed medium is the ideal solution, which has the same acoustic and electric parameters as the brain tissue.

The power attenuation mechanisms of microwave and acoustic waves are different and are discussed separately in the following sections.

### 2.1. Ultrasonic Wave Power Loss

The ultrasonic wave transmission loss is analyzed by Equation (2), in which p_z_ is the pressure amplitude at a distance of *z* meters from the source, *p*_0_ is the sound pressure of the source, and *α_u_* is the ultrasonic attenuation coefficient of the background medium. On referring to the *α_u_* values in Table 3, it decreases from air mineral oil to water, and there is almost no attenuation in water. Therefore, the ultrasonic wave propagation loss is the lowest in water.
(2)$$p_{z}=p_{0}e^{-\alpha_{u}z}$$

The reflection loss of the ultrasonic wave on the interface of the tissue is qualitatively analyzed by the acoustic impedance difference |ΔZ| between the tissue and the background medium. Reflection occurs when |ΔZ| is bigger than 0.1%, and larger |ΔZ| means stronger reflection and higher power loss. Acoustic impedance Z of a material is calculated by Equation (3), in which P is the density, and C is the sound speed in the material. The |ΔZ| of the four tissues in Table 3 has been calculated, and the results are shown in Figure 2a.
(3)Z=P×C

When it is the mineral oil, except for the breast fat, the values of |ΔZ| for brain, skull and muscle are the largest, indicating the highest acoustic power losses when compared with the other three background media. Since the acoustic parameters of the customed medium is the same with the brain tissue, their impedance difference is zero, indicating no acoustic power loss. The results with the customed medium for the other three tissues are the opposite with the mineral oil case. The |ΔZ| has the largest value for breast fat and has the smallest values for skull and muscle. The acoustic parameters of water and milk are similar, and the results are close for all kinds of tissues. Except for the ideal situation of the customed medium and the brain tissue, the values of |ΔZ| are relatively smaller for all the four tissues.

### 2.2. Microwave Power Loss

Under the hypothesis above, the microwave power loss occurs only in two ways: the transmission loss *P_L_* in the background medium and the reflection loss *R_L_* on the interface of the tissue. Therefore, the total power loss L is calculated by Equation (4).
(4)$$L=P_{L}+R_{L}$$

To calculate the PL in the background medium, the incident signal is considered a plane electromagnetic wave, where the electric field is given by Equation (5):(5)$$\vec{E}(z,t)=\vec{e_{x}}E_{xm}e^{-az}e^{j(\omega t-\beta z)}$$

In (5), $\vec{E}(z,t)$ is the electric field intensity vector along the direction of propagation, $\vec{e_{x}}$ is the unit vector in the *x* direction, *E_xm_* is the amplitude of the electric field, α is the microwave attenuation coefficient, *β* is phase constant, z is the transmission distance, *ω* is the angular frequency. The related formulas of α and *β* are shown in Equations (6) and (7), in which μ is magnetic permeability and is set to be the same as that of vacuum for all of the materials in Table 3. Therefore, the PL is calculated using Equation (8), which is derived from (5).
(6)$$\alpha =\omega \sqrt{\frac{\mu \varepsilon }{2}\left[\sqrt{1+{\left(\frac{\sigma }{\omega \varepsilon }\right)}^{2}}-1\right]}$$
(7)$$\beta =\omega \sqrt{\frac{\mu \varepsilon }{2}\left[\sqrt{1+{\left(\frac{\sigma }{\omega \varepsilon }\right)}^{2}}+1\right]}$$
(8)$$P_{L}(dB)=10\log(e^{2az})$$

The microwave return loss *R_L_* on the interface of the tissue is calculated by Equations (9)–(11), in which Γ is the voltage reflection coefficient.
(9)$$R_{L}(dB)=-20\log\left|\Gamma \right|$$
(10)$$\Gamma =\frac{\eta_{2}-\eta_{1}}{\eta_{2}+\eta_{1}}$$
(11)$$\eta =\sqrt{\frac{\mu }{\varepsilon }}$$

The total microwave power losses of the four tissues in the four kinds of background media are calculated, and the results are shown in Figure 2b. The value of distance z between the antenna and the tissue makes no difference for the purpose of comparison between different background media, which is set to be 100 mm in this study. According to Figure 2b, the total microwave power loss changes with the medium and the tissue. By definition, the value of L is negatively correlated to the total power loss. In other words, the bigger the value of L, the less the total power loss.

The result shows that the total microwave power loss is the least with the customed medium for the four tissues. Moreover, there is no reflection loss in this ideal condition and the total loss towards infinity. Except for the ideal background medium, milk can provide less total power losses for the brain, skull, and muscle, and after it, water. Mineral oil provides similar total power loss with water and the most for the other three tissues.

Above all, the background medium has a great influence on both microwave and ultrasonic power losses. The customed medium provides more satisfactory results in most cases. However, it is the ideal solution that needs special preparation and is hard to achieve. Milk has similar results to water. Considering the cleanliness and the cost, this work selects water as the background medium. Moreover, water has some other advantages in medical applications, such as a lack of prior preparation, higher comfort, and acceptance by the patients.

## 3. Antenna Design

### 3.1. Antenna Structure

It is easier to obtain the UWB characteristic by choosing the specific structures that feature ultra-wide bandwidth, such as spirals, cones, and multiple deformations based on the first two shapes. On the other hand, the antenna is supposed to be inserted into the tank from the bottom of the MITAT system and radiating along the vertically upward direction in order to cover the testing tissue. In this work, we designed a planar bifilar Archimedean spiral antenna to meet the requirements.

The proposed spiral radiating element is a balanced structure and wants equal currents along the two arms. However, the fed coaxial cable does not necessarily meet the requirement directly since some of the current may travel down the outside of the outer coax and lead to unbalanced operation. Therefore, the balun is designed to solve this problem and achieve impedance matching simultaneously. Since the feeding port of the spiral antenna is in the middle, the balun is designed to be connected with it from the vertical direction.

The overall structure and the perspective view of the proposed antenna are shown in Figure 3. Viewing from the outside, it is a cube with a coaxial port at the bottom. The dimensions of the cube are 55 mm × 55 mm × 60 mm. The antenna is made up of four parts: the radiating element, the balun, the shelter, and the reflector.

The reflector is a copper-clad board that is the same size as the radiating element. It reflects the radiation side lobes of the radiating element to concentrate the radiation beams and improves the directivity.

The shelter is a dielectric cavity that wraps up the balun between the radiating element and the reflector. It keeps the balun and the inner sides of the radiating element and the reflector free of water to avoid electric leakage and rust. Its dielectric constant is 6, which is close to the substrates of the microstrip circuits. Without the shelter, the electric conduction of water may cause higher power loss, and the great difference in dielectric constant between water and the substrate brings more difficulties to the work of impedance matching.

Both of the radiating element and the balun are designed with microstrip lines on the substrate with a permittivity of 4 and thickness of 1.6 mm. The specific design processes are described in the following sections.

### 3.2. Radiating Element Design

The front view of the proposed radiating element is shown in Figure 4a, while there is no metal layer coated on the other side of the substrate.

The two spirals constitute one kind of bifilar Archimedean spiral antenna, which is obtained by rotating the two arms of a dipole in the same direction under the Equation of the Archimedean spiral, as shown in Equation (12).
(12)$$r=r_{0}+k(\varphi -\varphi_{0})$$
(13)c=λf

Referring to Figure 4a, r is the distance from center point O to an arbitrary point at the spiral, r_0_ is the distance from point O to the starting point of either arm, *φ* is the azimuth angle, *φ_0_* is the starting angle, k is the growth rate of the spiral. The key to efficient radiation is to make currents on the two arms in-phase stack. In order to achieve this, r should be λ/2π, and λ is the wavelength. Points P and Q are two symmetrical points on the two spiral lines, respectively, and $\bar{OP}=\bar{OQ}$. The input currents at the starting points of the two arms are equal and opposite, namely, the phase difference between points P and Q is π. The current path from point Q to P’ is approximately equal to a semicircle arc with a radius of r. As a result, the current phase difference between point P and P’ is Δ*φ* = *π* + (2*π*/*λ*)*πr*. When Δ*φ* = 2*π*, namely, *r* = *λ*/2*π*, the radiations of the two arms realize an in-phase stack.

Above all, the main radiating area is the spiral ring with radius *r* = *λ*/2*π*. Since *λ* is inversely proportional to frequency *f*, the range of *r* corresponds to the range of working frequency. The ring number should be at least more than 1 in order to obtain a proper variation range of r. In addition, the size of the radiating element is supposed to be small in order to better integrate with the system. Therefore, the ring number is set to be 1.5. For the proposed antenna, r ranges from r_0_ to r_max_, which is 2 to 21 mm, which corresponds to a theoretical frequency range of 0.26 to 2.71 GHz. The relationship between *λ* and *f* is shown in Equation (13), in which c is the light speed in the vacuum. The S_11_ absolute bandwidth is 2.45 GHz, and the relative bandwidth is about 164.98%, which achieves ultra-wideband characteristics.

The polarization mode of the proposed antenna is defined mainly by the shape of the radiating element. The currents flow along the spirals and excite the electric field. The end of the electric field vector travels in circular or elliptical motion. In the rigorous definition of circular polarization, the vector track of the electric field must be circular. However, in practice, the axial ratio is used to define the polarization mode. The polarized ellipse of the electric field vector is shown in Figure 5. E is the instantaneous electric field vector, and EX and Ey are the transient components along the x and y directions, respectively. Axial ratio R is defined as Equation (14). Normally, when it is less than 2, the antenna can be considered circularly polarized. The axial ratios of the proposed antenna at different frequencies will be discussed in the related section.
(14)$$R=\frac{OA}{OB}$$

The growth rate k determines the value of r and thereby influences the radiation condition. Taking one spiral as an example, when the total length is a constant, *k* is negatively correlated with the number of spiral rings. A small *k* indicates more rings in the main radiation area, which leads to stronger radiation capacity and a smooth radiation pattern. However, this causes additional transmission loss and lower gain. On the other hand, large *k* means lower radiation capacity, which leads to an increase in terminal reflection at the end of the spiral and a deterioration of the circular polarization characteristic of the antenna. In this work, *k* is 0.022.

The arm width W and arm spacing S are related to the input impedance of the proposed radiating element. Normally, more stable impedance matching conditions are achieved by making equal the values of W and S, and they are usually to be set between 0.007*λ* to 0.01*λ* if they radiate in the air as common applications. However, the wavelength underwater is shorter, and the calculated values of W and S are too small, which may cause more insertion loss and difficulties in fabrication. The wavelength underwater is calculated by Equations (15) and (16). *β* is the phase constant. In this work, W is 1.5 mm, and S is 5.5 mm.
(15)$$\lambda =\frac{2\pi }{\beta }$$
(16)$$\beta =\sqrt{\pi f\mu \sigma }$$

The relative position of the starting points P1 and P2 of the two spiral arms has a significant influence on the impedance matching condition, as shown in Figure 4b. The rectangle L is a section of the balun, and its dimensions are 15 mm by 1.6 mm. P1 and P2 are connected to the balun and are symmetrically located on the two sides of L. The distance in the x direction between them is 4 mm. The simulated S_11_ versus the distance is shown in Figure 6. The optimal result is obtained when it is 4 mm. With this value, the best matching point is at 1.1 GHz, and the operation bandwidth is the widest.

### 3.3. Balun Design

The proposed balun and the dimensional information are shown in Figure 7. The width of L1, L2, and L6 is 3 mm. The shape of L3 is a trapezoid, and the top and bottom lengths are 3.6 mm and 3 mm, respectively. The gap D1 under L4 is an isosceles trapezoid, and the upper length is 0.2 mm.

The theoretical total length of the balun is a quarter wavelength of 1.1 GHz, while there is a small amount of change in simulation and optimization. According to Equation (17), the input impedance Z_IN_ of a quarter wavelength microstrip line is defined by the characteristic impedance Z_0_ and the load Z_L_. Z_0_ is closely related to the microstrip line width. Therefore, the input impedance of the balun is optimized mainly by changing the line widths.
(17)$$Z_{IN}=\frac{Z_{0}^{2}}{Z_{L}}$$

The proposed balun comprises three units: U_1_, U_2_, and U_3_. U_1_ is the microwave input unit and is designed to be a coplanar waveguide, which is balanced by nature and can force the unbalanced fed coaxial line to be balanced. The source power is input from L_1_, and the two metal squares on both sides of L_1_ at the bottom are connected to the ground by two vials coated with a metal conductor. To achieve good grounding conditions, the metal area of L_4_ on the other side of the substrate is larger. The gap D_1_ under L_4_ helps to add more variables for easy optimization. U_2_ is the transition unit, which contains L_2_ and L_5_. L_5_ is a tapered line that is introduced to gradually convert L_4_ to L_6_ in case the impedance mismatch problem is caused by the large width difference between the two lines. U_3_ is the linking-up unit that connects the balun with the spiral radiating element. The upper ends of L_3_ and L_6_ tilt to the left to connect with points P_1_ and P_2_ in Figure 4b.

According to the transmission line theory, the microstrip line with a quarter wavelength converts the input impedance by Equation (17), in which Z_L_ is the load impedance, Z_0_ is the characteristic impedance, and Z_IN_ is the input impedance. Z_0_ can be calculated theoretically by Equations (18) and (19), in which W is the width of the microstrip line, h is the thickness of the substrate, η_0_ is the wave impedance in free space, ε_eff_ is the equivalent relative dielectric constant which is calculated by Equation (20) to (22). Therefore, Z_0_ is closely related to the width of the microstrip line and plays a key role in S_11_ improvement. In order to reduce the insertion loss caused by the width differences, the widths of L1, L2, L3, and L4 in Figure 7 are set to be the same.
(18)$$Z_{0}=\frac{\eta_{0}}{2\pi \sqrt{\varepsilon_{\mathrm{eff}}}}In\left[\frac{Fh}{W}+\sqrt{1+{\left(\frac{2h}{W}\right)}^{2}}\right]$$
(19)$$F=6+\left(2\pi -6\right)\cdot \exp\left[-{\left(\frac{30.666h}{W}\right)}^{0.7528}\right]$$
(20)$$\varepsilon_{\mathrm{eff}}=\frac{\varepsilon_{r}+1}{2}+\frac{\varepsilon_{r}-1}{2}{\left(1+\frac{10}{\mu }\right)}^{-ab}$$
(21)$$a=1+\frac{1}{49}In\left(\frac{\mu^{4}+{\left(\frac{\mu }{52}\right)}^{2}}{\mu^{4}+0.432}\right)+\frac{1}{18.7}In\left(1+{\left(\frac{\mu }{18.1}\right)}^{3}\right)$$
(22)$$b=0.564{\left(\frac{\varepsilon_{r}-0.9}{\varepsilon_{r}+3}\right)}^{0.053}$$

The simulated S_11_ versus the width of L1 are shown in Figure 8. The best matching point is at 1.1 GHz when it is 3 mm. Different from the distance between points P1 and P2, this parameter mainly influences the best matching point and has a very small impact on the bandwidth. Except for 3.6 mm, the bandwidths with other values all vary from approximately 0.8 GHz to 2.8 GHz.

Since the wavelength in water is shorter, which calls for more cells in the meshing, the simulation of the proposed water immersion antenna is time-consuming. Moreover, parameter sweeping requires more memory of the computer. For the confined condition, the results of Figure 6 and Figure 8 are simulated with coarse node spacing.

## 4. Antenna Performance

### 4.1. Antenna Simulation

To illustrate the proposed antenna’s working performance, the S_11_, axial ratios, and radiation patterns at different frequencies have been simulated and analyzed.

The simulated results of S_11_ are shown in Figure 9. The 10 dB impedance bandwidth is 1.91 GHz, from 0.84 to 2.75 GHz. This fits the theoretical result analyzed in the previous section, which is from 0.26 to 2.71 GHz. The subtle variation between them is due to the truncation effect of the radiating element. The terminals of the two spiral lines are truncated, and the small amounts of current that are not radiated as expected are reflected at the terminals, which causes destruction to the traveling wave state of the currents flowing along the spiral lines. According to the simulated results, the influence of the truncation effect is very small, and the proposed antenna performs well in impedance matching.

The characteristic of circular polarization is described by the axial ratio. The average axial ratios at different frequencies of the proposed antenna are shown in Figure 10. It is less than 2 from 0.91 to 3.97 GHz, indicating the good and stable circular polarization characteristic in an ultra-wideband of the proposed antenna.

The radiation patterns at six sample frequencies are shown in Figure 11. For all frequencies, the main lobes point to the direction z accurately, and there are very few side lobes. The radiation pattern bandwidth of the proposed antenna is from 0.6 to 1.8 GHz. When the frequency is over this range, the main lobe deviates to the side of more than 10 degrees, and the directivity may decrease. The directivities and 3 dB angular widths at the sample frequencies are listed in Table 4. With the increase in frequency, the directivity increases, and the 3 dB angular width decreases. This is due to the corresponding change in the electric dimension of the balun. From 0.9 to 1.8 GHz, the directivity is greater than 11 dBi.

The 3 dB beam angle reflects the microwave beam focusing condition, which is related to the microwave illumination performance in the MITAT system. Narrow angle means strong beam focusing, which may lead to imperfect power coverage on the target tissue and cause damage. A wide beam angle makes it easy to provide full cover, but the illumination power may be insufficient. On the other hand, the testing target is put in the far field area of the antenna in order to achieve stable radiation illumination, which is R meters away and is calculated by Equation (23). L is the largest dimension of the antenna, *λ* is the wavelength underwater. As an example, the calculated R is 70.6 mm at 1.1 GHz for the proposed antenna. The 3 dB beam angle is 45 degrees, and it covers a circle area with a radius of 82.8 mm at a distance of 100 mm. If the cross-sectional area of the testing target is larger than this, the illumination performance is not satisfied, and the power absorption distribution cannot reflect the real substance distribution. This is validated by the simulation of the effective microwave absorption area of brain tissue in the later section. However, the relative position of the antenna and the target tissue is adjusted to obtain the best microwave illumination condition during the experiment.
(23)$$R=\frac{2L^{2}}{\lambda }$$

Above all, the frequency bandwidths defined by impedance matching, axial ratio, and radiation pattern of the proposed antenna are calculated in Table 5, and the antenna is an ultra-wideband in all three aspects.

### 4.2. Antenna Fabrication

The fabricated antenna is shown in Figure 12. The radiating element and the balun are fabricated on the FR4 substrates, both with permittivity of 4 and thickness of 1.6 mm. The black object is the shelter, and its material is epoxy resin, with a permittivity of approximately 6. To achieve leak-tight and tight wrapping performance, a cube-shaped cavity mold is printed by a 3D printer in advance. The inner circuit is put in the mold, and the epoxy resin liquid is injected into it. After 12 h, the liquid is solidified, and the mold is easy to take off.

### 4.3. Antenna Measurement

The diagram of the testing system is shown in Figure 13. The microwave vector network analyzer is used to measure the S_11_. The tank is plastic and used to retain water. During the testing, the external surface of the reflector of the proposed antenna must stay free of water in case of electrical leakage from the feed port. The other parts of the proposed antenna should be fully immersed in water. The distances from the antenna to the tank walls should be larger than the far field distance R in order to reduce the negative effects in impedance matching and radiation performance caused by the reflected power at the walls. In this study, the tank dimensions are 75 cm × 60 cm × 45 cm. In order to ensure compliance with the far field requirement, the antenna is held in the center of the cross-section of the tank during the testing, and the water depth is 40 cm.

The measured S_11_ is presented in Figure 9, showing good agreement with the simulated result at frequencies from 0 to 1.87 GHz. The minor differences between them are due to the increased radiation losses in testing, the insertion loss caused by errors in the manufacturing process, and the parasitic effects of discontinuous components. At frequencies above 1.87 GHz, the measured data fluctuations are higher, indicating that the insertion loss becomes higher at high frequencies. The measured 10 dB frequency bandwidth is 2 GHz, corresponding to the frequency range of 0.9 to 2.9 GHz.

For the confined condition, the radiation pattern of the proposed antenna is not measured. However, the radiation performance is proven from the side by analyzing the microwave power absorption distribution of the target tissue. This is studied in the following section.

## 5. Microwave Power Absorption under the Illumination of the Proposed Antenna

As described in the introduction, the microwave power absorption distribution reflects the substance distribution of the testing target. The high degree of conformity between the two implies good microwave illumination from the antenna. In this study, we build models to simulate the microwave power absorptions of two tissue phantoms under illumination from the proposed antenna. The models and phantoms are shown in Figure 14.

The input microwave power is 50 kW at 1.1 GHz. The pulse frequency is 10 Hz with a pulse width of 0.5 μs. The distance between the antenna and the phantom is 100 mm. Theoretically, the 3 dB power beam of the proposed antenna covers a circular area of 82.8 mm in diameter at this distance. Phantom 1 is the model of the cylindrical brain tissue; the diameter is 60 mm, and the height is 2 mm; the electrical parameters are listed in Table 3. The 2-dimensional bird’s eye view of the microwave absorption distribution is shown in Figure 15, and the energy value per cubic meter is represented in different colors.

The maximum amount of absorbed power is in the center and gradually decreases to half at the edge of the phantom, as shown in the two markers in Figure 15. This distribution pattern conforms to the radiation pattern of the proposed antenna, and the resulting uneven molecular thermal motions further induce the mild uneven distribution on the same radius. Moreover, the reflections at the boundaries cause more apparent fluctuations near the phantom edge. However, highly uneven distribution will decrease the accuracy of the blur identification and the quality of the restored image. According to Figure 15, the effective microwave absorption area of phantom 1 is defined as a circle 40 mm in diameter.

Phantom 2 in Figure 14b is the model in which a cylindrical tumor is inserted in the cubic brain tissue. The dimensions of the cross-section of the cubic are 30 mm × 30 mm, and the height varies with that of the cylinder. The relative permittivity of the tumor is close to the muscle and is set to be 70; the conductivity is 2 S/m. Set the phantom height to 2 mm and change the tumor diameter from 4 mm, 10 mm, 20 mm, and 30 mm to 40 mm; the related 2-dimensional bird’s eye views of the microwave absorption distributions are shown in Figure 16. When the tumor diameter is 4 mm, the outline of the tumor has a slight deformation, and the microwave absorption distribution of the brain tissue near the tumor is nonuniform. When the tumor diameter is larger, the outline of the tumor is clear and smooth, with almost no deformation. Moreover, the power absorption distribution inside the tumor is uniform. In this case, the contrast of the tumor and brain tissue is significant. The minor fluctuations near the edge of the brain tissue are caused by the boundary effect and the slight inhomogeneous illumination of the antenna. This only appears at the boundary, and the power intensity is much less than in the tumor; therefore, it makes no difference to the tumor image identification.
(24)$$D={\left(\omega \sqrt{\frac{\mu \varepsilon }{2}(\sqrt{1+{\left(\frac{\sigma }{\omega \varepsilon }\right)}^{2}}-1)}\right)}^{-1}$$

The imaging depth is also studied. Theoretically, the microwave penetrating depth D is calculated by Equation (24) [61], *ω* is the angular frequency, *μ* is magnetic permeability, *ε* is the dielectric constant, *σ* is conductivity. Therefore, the calculated microwave penetrating depth D in the tumor is 32 mm. Due to the difference in dielectric properties between the tumor and the surrounding healthy tissue and the stray loss, the effective penetrating depth for a high-quality image should be less than the calculated value. The simulated results are shown in Figure 17. The diameter of the phantom is 10 mm. When the height is 2 mm, the tumor outline is clear, and the energy distribution inside the tumor is uniform. When the height exceeds 5 mm, more energy gathers in the middle, and two small sidelobes appear at the edge of the tumor. Further increasing the height, the sidelobes become bigger, and the total amount of microwave absorption decreases. When the height comes to 10 mm, the absorbed energy at the center of the tumor is less by a third compared with the 2 mm case. It is even worse for the 15 mm case; though the tumor is still recognizable, the contrast becomes very low.

Above all, under the microwave illumination provided by the proposed antenna under water, brain tissue and tumors within the dimensions of 5 mm in thickness and 40 mm in diameter absorb microwave power sufficiently and uniformly, which induces strong ultrasonic signals. When the tumor is inside brain tissue, there is a huge difference in absorbed microwave power between both, and the microwave absorption distribution map is in high contrast and accurately reflects the fact. Assuming the ultrasonic part of the MITAT system is ideal, the final images should have no difference from the microwave absorption distribution map, indicating the availability and potentiality of the proposed antenna in brain tumor detection.

## 6. Conclusions

In this article, an ultra-wideband water immersion antenna for underwater thermoacoustic sensing in biomedical imaging applications has been designed. The power losses of microwave and ultrasonic waves in water and mineral oil of the MITAT system have been analyzed and compared. Theoretical analysis, simulations, and experiments have been made to analyze and verify the good working performance of the proposed antenna. As a result, it has ultra-wide bandwidth in impedance matching, axial ratio, and radiation pattern, indicating stability, flexibility, and reliability in biomedical applications. The simulations on microwave absorption distribution of brain tissue and tumor have been made; the results indicate that the proposed antenna provides satisfactory illumination and thereby achieves uniform microwave absorption distribution inside the tumor and high contrast between the tumor and the surrounding healthy brain tissues. In general, this work proves that the proposed antenna has the advantages of ultra-wideband, stable performance, lightweight, antirust, and low cost and that it works cooperatively with the ultrasonic sensor to make the MITAT efficient in biomedical imaging. This work would also provide a basis for further development and optimization of water immersion antennas for various applications, such as underwater sensor networks and underwater communications. With the development of dielectric materials, more flexible materials could be applied in the antenna design, such as artificial materials and specific thin or soft substrate.

## Figures and Tables

**Figure 1 sensors-24-06311-f001:**
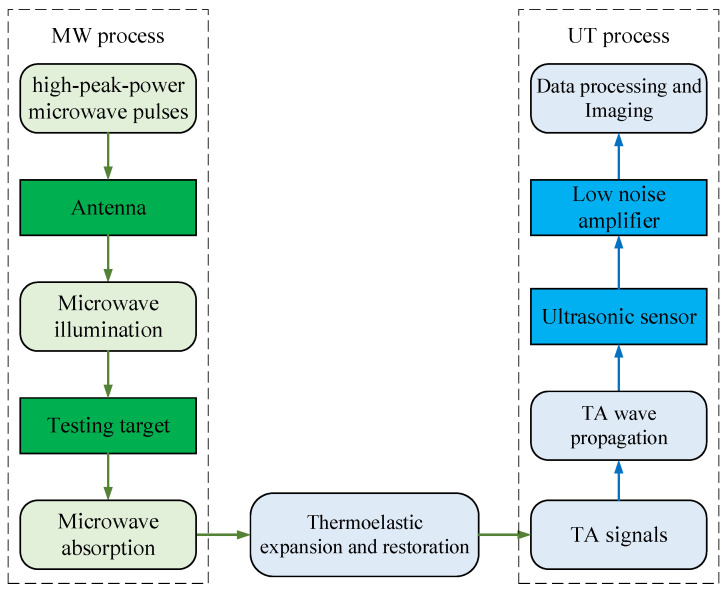
Block diagram of the process of MITAT.

**Figure 2 sensors-24-06311-f002:**
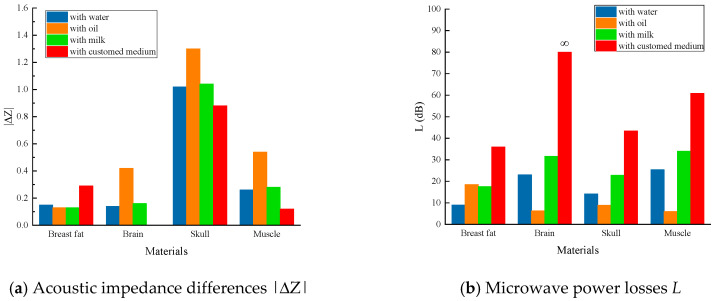
Histograms of acoustic impedance differences and microwave power losses when the four tissues are illuminated in mineral oil and water, respectively.

**Figure 3 sensors-24-06311-f003:**
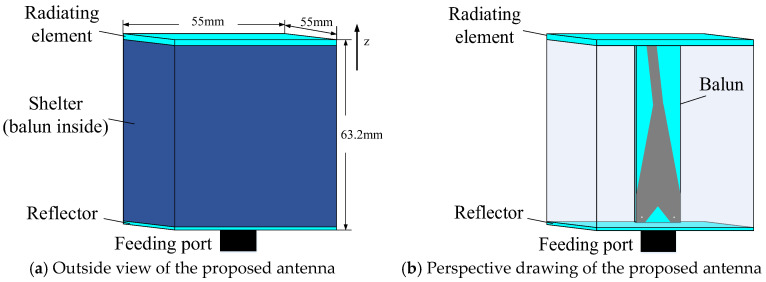
Schema of the decomposed structures for the proposed antenna.

**Figure 4 sensors-24-06311-f004:**
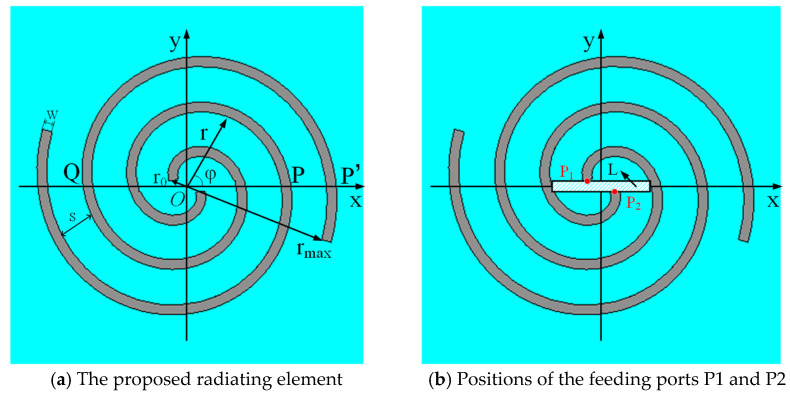
Front views of the proposed radiating element.

**Figure 5 sensors-24-06311-f005:**
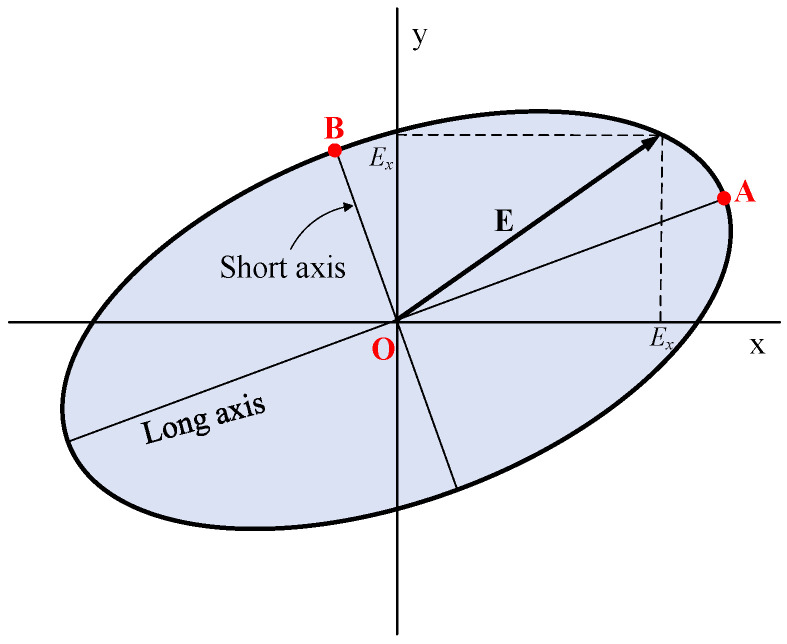
F The polarized ellipse and the transient components Ex and Ey of the electric field vector E.

**Figure 6 sensors-24-06311-f006:**
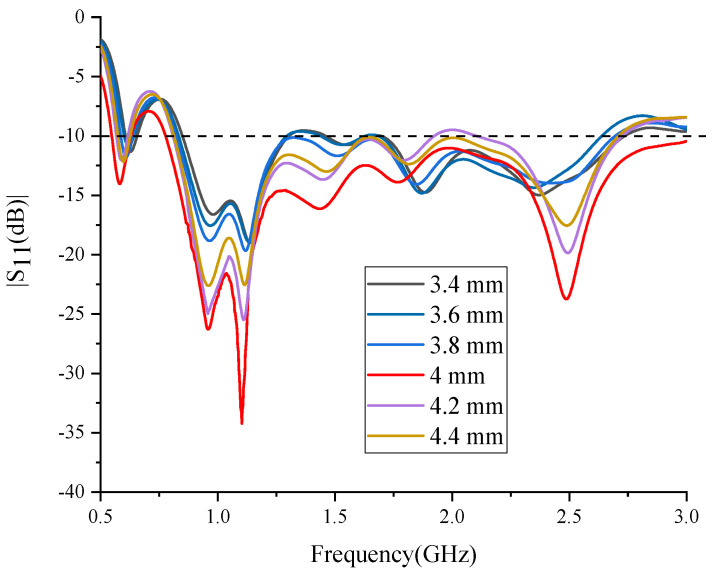
The simulated S_11_ versus the distance between points P1 and P2.

**Figure 7 sensors-24-06311-f007:**
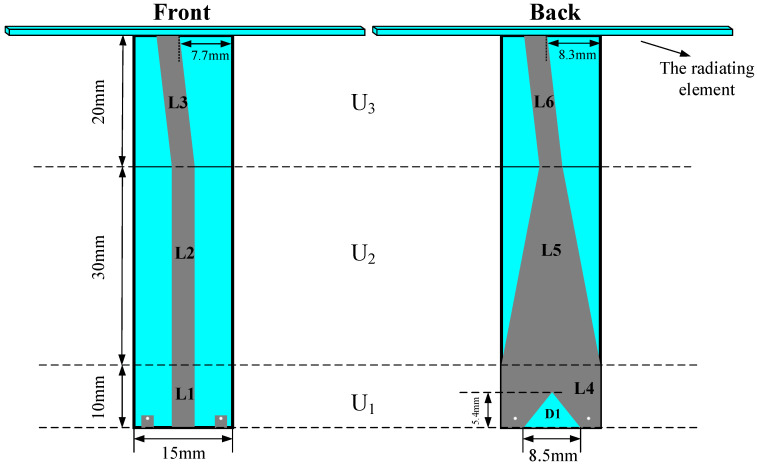
Front and back views of the proposed balun.

**Figure 8 sensors-24-06311-f008:**
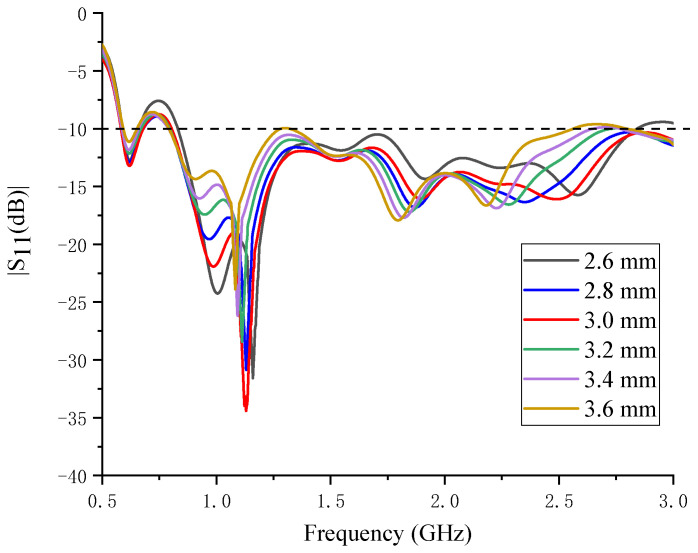
The simulated S_11_ versus the width of L1.

**Figure 9 sensors-24-06311-f009:**
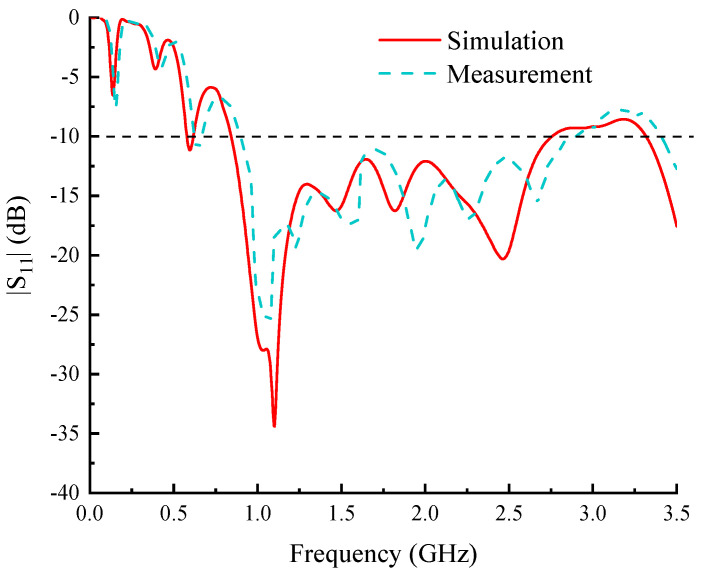
Parameters of the proposed antenna.

**Figure 10 sensors-24-06311-f010:**
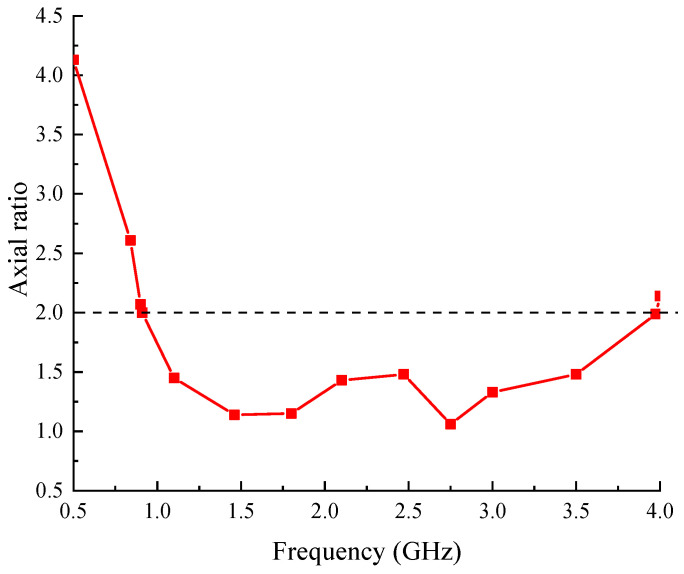
Axial ratios of the proposed antenna.

**Figure 11 sensors-24-06311-f011:**
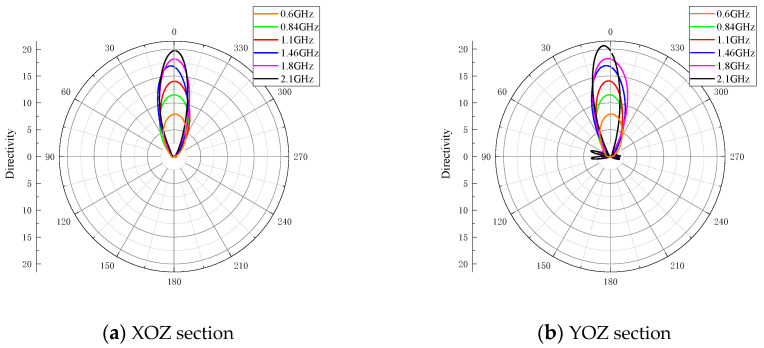
Radiation patterns of XOZ and YOZ sections of the proposed antenna.

**Figure 12 sensors-24-06311-f012:**
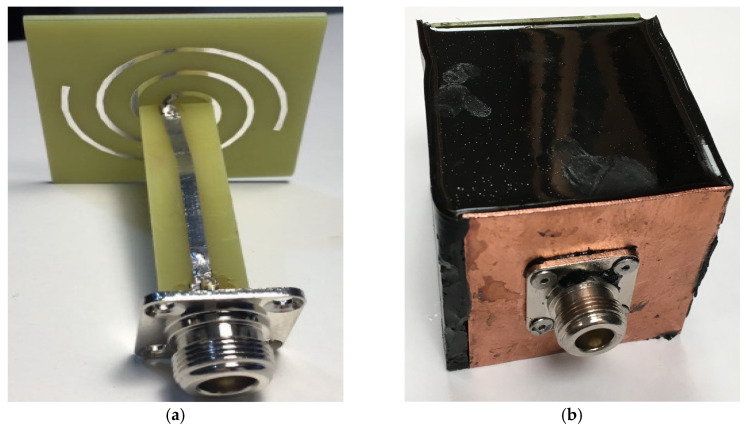
Photographs of the proposed antenna: (**a**) is the inner circuits that contain the fabricated radiation element and balun, (**b**) is the full view of the fabricated antenna, and the black object is the shelter.

**Figure 13 sensors-24-06311-f013:**
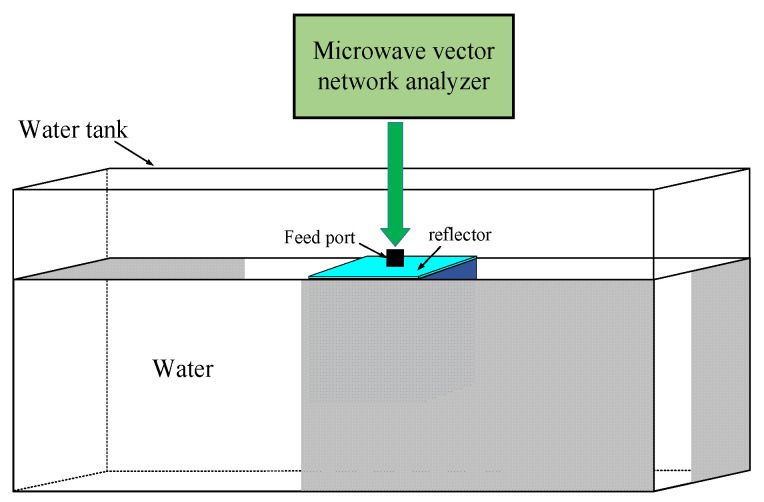
Diagram of the testing system.

**Figure 14 sensors-24-06311-f014:**
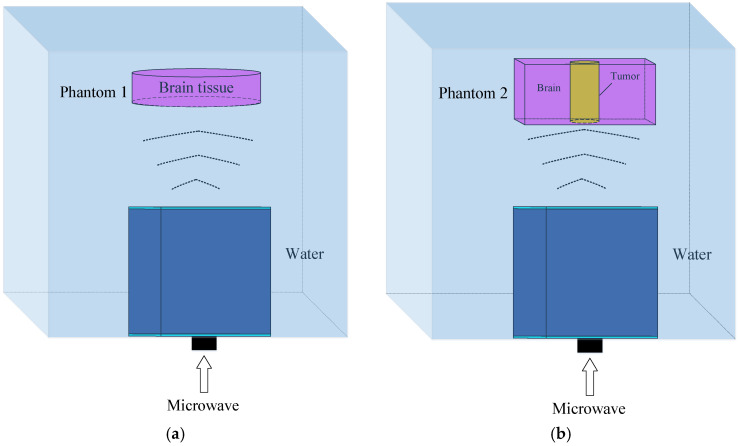
Diagrams of the microwave absorption simulation models of the phantoms, (**a**) phantom 1; (**b**) phantom 2.

**Figure 15 sensors-24-06311-f015:**
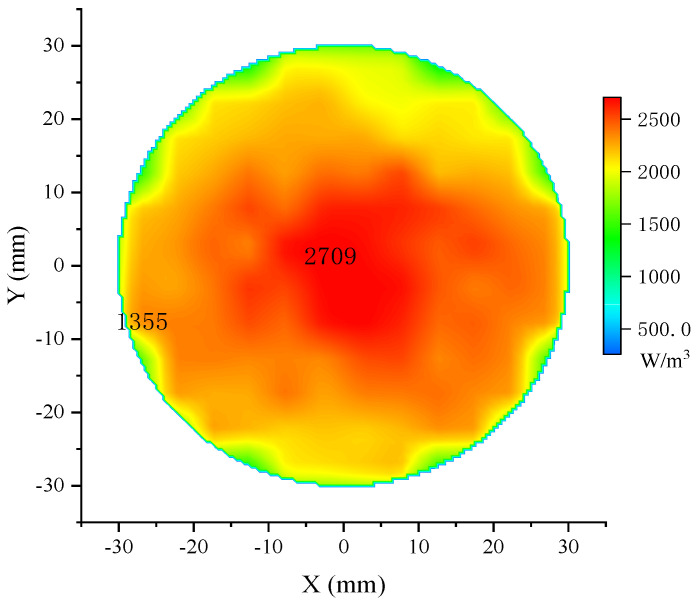
Microwave absorption distribution map of phantom 1.

**Figure 16 sensors-24-06311-f016:**
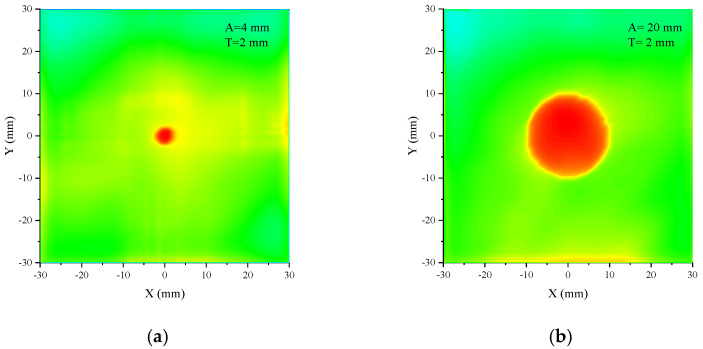
The microwave absorption distribution maps of phantom 2 with different tumor diameters. A is tumor diameter, and T is phantom height.

**Figure 17 sensors-24-06311-f017:**
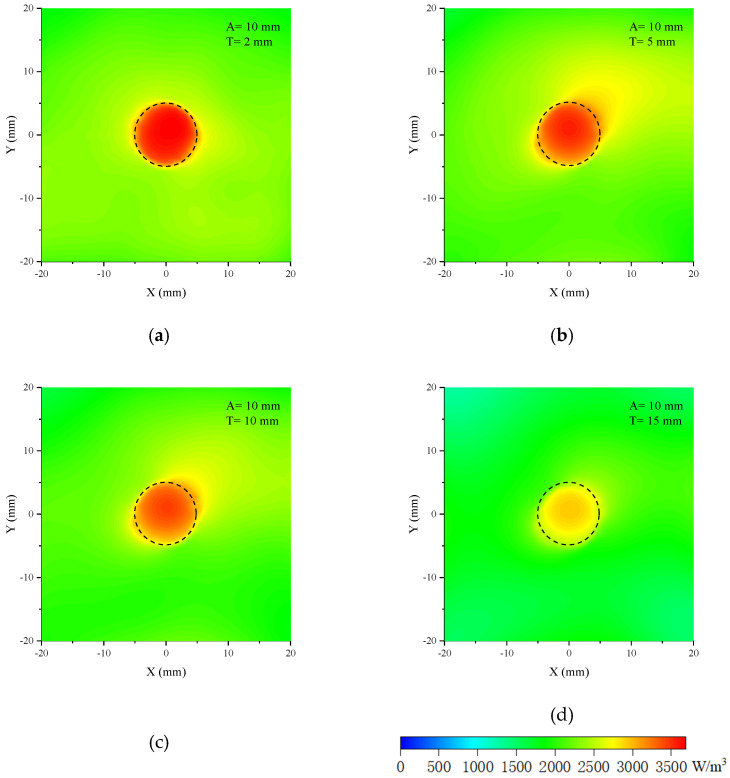
The microwave power absorption distribution maps of phantom 2 with different phantom heights. A is the tumor diameter, and T is the phantom height.

**Table 1 sensors-24-06311-t001:** Comparison of novel antennas as developed for MITAT.

Reference	Antenna	Features	Application Scenario	Background Medium
[21]	Quasi-conical spiral antenna	1. Large image domain 2. Homogeneous 3. Circular polarization	Breast and prostate imaging	Mineral oil
[22]	Helical antenna	1. More homogenous illumination 2. Circular polarization	Cylindrical tumor phantom	Mineral oil
[22]	Horn antenna	1. Less homogenous illumination 2. Linear polarization	Cylindrical tumor phantom	Mineral oil
[23]	Polarized antenna	1. High SNR 2. Homogenous illumination 3. Circular polarization	Phantoms of breast tumor imaging	Mineral oil
[24]	Waveguide	1. Quantitatively reconstruct dielectric properties of biological samples 2. A 5% experimental reconstruction error	Reconstruct inhomogeneous biological samples	Acoustic coupling
[25]	Cascaded Waveguide	1. Two waveguides in tandem 2. The choice of optimal microwave frequency based on SNR is broad for malignant tumors 3. Using gain compensation to counteract the microwave attenuation and improve the image contrast	Fat tissue and muscle tissue sample imaging	Water
[26]	Dipole antenna	1. Portable 2. Anti-Gal1-Fe_3_O_4_ nanoparticles through intravenous infusion is needed 3. Can identify tiny pancreatic tumors	Tiny pancreatic tumors imaging	Mineral oil
[27]	Beamforming array antenna	1. The peak power can be reduced 2. The focus point can be controlled 3. Wearable	Breast cancer imaging	Mineral oil
This work	Water-immersion antenna	1. Water immersion and antirust 2. Homogenous illumination 3. Circular polarization 4. Ultra-wideband	Brain and tumor tissues imaging	Water

**Table 3 sensors-24-06311-t003:** Acoustic and electric parameters of different materials [59,60].

Category	Material	Density kg/m^3^	Sound Speed m/s	Acoustic Impedance 10^6^ kg/m^2^⋅s	α_u_ * Np/m	ε_r_ *	σ * S/m
Background medium	Air	1	330	0.00033	13.81	1	0
	Mineral oil	825	1440	1.19	3.63	3	0.02
	Water	994	1480	1.47	0.02	78	0.23
	milk	945	1535	1.45	40	78	0.7
	Customed medium	1046	1540	1.61	10.36	48.9	1.31
Target tissue	Breast fat	911	1450	1.32	4.36	5.41	0.1
	Skull cancellous	1178	2117	2.49	47	20.6	0.364
	Brain	1046	1540	1.61	10.36	48.9	1.31
	Muscle	1090	1590	1.73	7.11	54.8	0.978

* α_u_ is the ultrasonic attenuation coefficient at 1 MHz; ε_r_ is the relative permittivity; and σ is the conductivity.

**Table 4 sensors-24-06311-t004:** Directivity and 3dB angular width at different frequencies.

Frequency (GHz)	Directivity (dBi)	3dB Angular Width (Degree)
0.6	8.98	62.3
0.84	10.6	53.9
0.9	11	51.3
0.91	11	50.8
1.1	11.5	45
1.46	12.3	37.9
1.8	12.6	34.8

**Table 5 sensors-24-06311-t005:** The bandwidths of impedance, axial ratio, and radiation pattern of the proposed antenna.

Category	Absolute Bandwidth (GHz)	Relative Bandwidth	Center Frequency (GHz)
Impedance	1.91	106.4%	1.795
Axial ratio	3.06	125.4%	2.44
Radiation pattern	1.2	100%	1.2

## Data Availability

The raw data supporting the conclusions of this article will be made available by the authors on request.

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
