# Peer review of "Design and Implementation of an Ultra-Wideband Water Immersion Antenna for Underwater Ultrasonic Sensing in Microwave-Induced Thermoacoustic Tomography"

_sensors, 2024, doi:10.3390/s24196311_

Round 1

Reviewer 1 Report

Comments and Suggestions for Authors

1. A comparison with related antennas for similar application scenarios should include.

2. In the introduction, there is a statement that “other materials may also be used as the background medium.” Could you provide some results using other background materials for comparison? For example, materials that more closely resemble the human body.

3. The relevant analysis and performance results of circular polarization need to be provided since this is a circularly polarized antenna.

4. Is the overall size of the antenna appropriate? Are there any further methods to reduce its size?

Comments on the Quality of English Language

Minor editing of English language required.

Reviewer 2 Report

Comments and Suggestions for Authors

In this paper, author present UWB antenna for ultrasonic sensing in microwave induced thermoacoustic tomography. The paper is well written and al results are well discussed. I have some suggestions.

1) How did author get this final UWB from this antenna? The author must follow some design steps to get UWB? Also show the formula to get optimized parameters of proposed geometry. In every antenna, some of the parameters play key role in S11 improvement, author can add its parametric results.

2) In literature, many antenna is proposed, which is simple to fabricate and offer good bandwidth. For example monopole antennas, patch antenna as given in single iterated fractal inspired UWB antenna with reconfigurable notch bands for compact electronics, and also Band enhancement of compact flexible antenna for wlan, wifi and c-band applications.

3) It is better to provide comparison table to compare outcomes (size, operational band, gain, design methodology, applications) of proposed antenna with other works. This will help to make novelty of work clear.

4) Selection of dielectric material is important RF sensors designing. The author chose for this proposed antenna and radiating element is also important. As in Flexible dielectric materials: potential and applications in antennas and RF sensors.

Comments on the Quality of English Language

Just revise paper to check grammar and typo mistakes. 

Round 2

Reviewer 1 Report

Comments and Suggestions for Authors

Thanks for the revision. I have no further comments.

Comments on the Quality of English Language

Minor editing of English language required.

Author Response

We appreciate the time and effort that you dedicated to providing feedback on our manuscript and are grateful for the insightful comments on and valuable improvements to our paper.

Thanks again!

Feifei Tan

Reviewer 2 Report

Comments and Suggestions for Authors

Comment 1, 2 and 4 is not addressed properly. Kindly check it again.

Thanks
